# Towards the Standardization of Mesenchymal Stem Cell Secretome-Derived Product Manufacturing for Tissue Regeneration

**DOI:** 10.3390/ijms241612594

**Published:** 2023-08-09

**Authors:** Batoul Chouaib, Mandana Haack-Sørensen, Franck Chaubron, Frederic Cuisinier, Pierre-Yves Collart-Dutilleul

**Affiliations:** 1LBN, University of Montpellier, 34000 Montpellier, France; batoul_chouaib@hotmail.com (B.C.); frederic.cuisinier@umontpellier.fr (F.C.); 2Human Health Department, IRSN, French Institute for Radiological Protection and Nuclear Safety, SERAMED, LRMed, 92262 Fontenay-aux-Roses, France; 3Cardiology Stem Cell Centre 9302, Rigshospitalet University of Copenhagen, Henrik Harpestrengsvej 4C, 2100 Copenhagen, Denmark; 4Institut Clinident BioPharma, Biopôle Clermont-Limagne, 63360 Saint Beauzire, France; fchaubron@institut-clinident.com; 5Faculty of Dentistry, University of Montpellier, 34000 Montpellier, France; 6Service Odontologie, CHU Montpellier, 34000 Montpellier, France

**Keywords:** mesenchymal stem cell, secretome, manufacturing, tissue regeneration

## Abstract

Mesenchymal stem cell secretome or conditioned medium (MSC-CM) is a combination of biomolecules and growth factors in cell culture growth medium, secreted by mesenchymal stem cells (MSCs), and the starting point of several derived products. MSC-CM and its derivatives could be applied after injuries and could mediate most of the beneficial regenerative effects of MSCs without the possible side effects of using MSCs themselves. However, before the clinical application of these promising biopharmaceuticals, several issues such as manufacturing protocols and quality control must be addressed. This review aims to underline the influence of the procedure for conditioned medium production on the quality of the secretome and its derivatives and highlights the questions considering cell sources and donors, cell expansion, cell passage number and confluency, conditioning period, cell culture medium, microenvironment cues, and secretome-derived product purification. A high degree of variability in MSC secretomes is revealed based on these parameters, confirming the need to standardize and optimize protocols. Understanding how bioprocessing and manufacturing conditions interact to determine the quantity, quality, and profile of MSC-CM is essential to the development of good manufacturing practice (GMP)-compliant procedures suitable for replacing mesenchymal stem cells in regenerative medicine.

## 1. Introduction

The regenerative effects promoted by mesenchymal stem cells (MSCs) are mainly associated with the secretion of bioactive molecules, which are endowed with paracrine activity, including soluble factors (proteins, nucleic acids, lipids) and extracellular vesicles (EVs) [1], and defined as the secretome or mesenchymal stem cell-conditioned medium (MSC-CM) [2,3]. The preparation of MSC-CM consists of leaving the cells in culture for a certain period, before using centrifugation to collect their growth medium containing all their secretions.

The number of peer-reviewed articles focusing on the use of MSC-CM has increased exponentially in the last few years [4,5]. MSC-CM is emerging as an alternative to direct MSC therapy and has a promising prospect to be produced as pharmaceuticals for regenerative medicine [4]. Indeed, the use of CM has several advantages compared to the use of stem cells. As it is devoid of cells, there is no need to match the donor and the recipient to avoid rejection problems [4], and the absence of replicating (allogeneic) cells in secretome fractions significantly improves patient safety profile.

The concentrations of factors contained in MSC-CM are relatively low. Thus, the use of MSC-CM does not induce any severe histological inflammatory responses as could be observed with the clinical use of recombinant factors [6]. The low metabolic activity allows more efficient quality controls and quality assurance. Moreover, CM can be manufactured, freeze-dried, packaged, and transported more easily than cells. The simplicity of storage provides the basis for cost-efficient shipping of this potentially therapeutic substance [7]. Although all studies converge around the regenerative potential of MSC-CM and its derivatives as the therapeutically active components of MSCs, the literature reveals high variability in terms of MSC sources and manufacturing processes. This highlights the challenges to the clinical translation of MSC-CM and its derivatives and underlines the importance of method and protocol standardization for MSC secretome-derived products of GMP grade [8,9,10,11,12,13,14]. Process optimization and standardization are needed to avoid changes in protein secretion profiles, properly compare studies, and ensure effective quality control [15]. Furthermore, understanding how cell sources, donors, and culture conditions interact to determine the quantity, quality, and types of biomolecules secreted by MSCs in conditioned medium and its derivatives is essential to the development of bioprocesses aimed at scaling up the production of secretome-derived products [11].

Secretome characterization is required to confirm the reproducibility of the manufacturing method and because of regulatory requirements concerning quality, safety, and efficacy. The tools mostly used for studying the expression of the MSC secretome in vitro include protein/peptide separation techniques followed by protein identification by mass spectrometry and immunological assays [16]. The characterization of the MSC-CM using antibody-based techniques such as Elisa and antibody arrays provides only a narrow window of factors secreted by MSCs. Global proteomic approaches would better clarify the potential complexity of MSC secretome, with the ultimate aim of obtaining clear and defined MSC-CM profiles for appropriate use of each MSC-CM according to its profile. However, all these techniques can play an essential role in the standardization of the production of MSC-CM and its derivatives, which are hampered by variations in MSC source, donors, cell expansion, cell passage number, conditioning period, cell culture medium, microenvironment cues, and secretome-derived product purification (Figure 1). Here, we discuss these different variations and illustrate their effect on the MSC secretome profile and protein concentrations using microarray analysis, protein electrophoresis, and the Bradford assay.

## 2. Cell Sources

The MSC secretome varies significantly according to the source tissue [17]. Baglio et al. demonstrated that stem cells from bone marrow (BMSCs) and adipose tissue (ASCs) secreted different tRNA species that may be relevant for clinical applications [18]. Pires et al. illustrated the difference in profiles and efficiencies of secretomes produced from BMSCs, umbilical cord mesenchymal stem cells (UMSCs), and ASCs. Although important changes were observed within the secretome of the cell populations that were analyzed, all cell populations shared the capability of secreting important regulatory molecules [19]. Du et al. demonstrated the heterogeneous proangiogenic properties of BMSCs, ASCs, UMSCs, and placental chorionic villi (PMSCs) and suggested that BMSCs and PMSCs might be preferred in clinical applications for therapeutic angiogenesis [20]. A study conducted by Kehl et al. compared the angiogenic potential of MSC secretomes from ASCs, BMSCs, and UMSCs and suggested UMSCs as the most potent MSC source for inflammation-mediated angiogenesis induction, while the potency of ASC secretomes was the lowest [21]. In contrast, Hsiao et al. suggested that ASCs may be the preferred source, after a comparison of the expression of angiogenic paracrine factors in ASCs, BMSCs, and MSCs from dermal tissues [22]. Ribeiro et al. also detected more factors in ASC-CM than in UMSC-CM [23]. Hsieh et al. suggested that UMSCs, because of their secreted factors involved in angiogenesis and neurogenesis, were better than BMSCs for promoting in vivo neuro-restoration and endothelium repair [24].

Taken together, all these studies indicate that MSC secretomes differ between cell sources. Thus, it is important, before the clinical translation of MSC secretome-based products, to determine which cellular sources provide the most potential for each application associated with tissue regeneration.

## 3. Donors

Mesenchymal stem cells are used as a starting material for CM manufacturing, and their properties crucially influence the composition of CM. Therefore it is necessary to standardize MSC recovery, and, also, to take into account the variability of MSC donor characteristics during manufacturing [14].

The effect of donor variability on the MSC-CM profile is currently not well understood. The composition of MSC-CM appears to be influenced by donor variability in some studies [25], as well as their effect in vitro, including opposite effects sometimes [17]. However, other studies suggested that the trophic nature of MSCs and their cytokine profiles do not depend on donor individuality [26]. They showed a similar set of proteins expressed in the MSC secretomes of two different donors, while donor-dependent variations were just reflected by the different expression levels of each protein [27].

Donor-related factors may be responsible for the impact of donor variability on MSC function and corresponding secretome. These characteristics are age, gender, metabolic state, and disease. Whilst very few studies have reported gender dimorphism of MSC effects [28], functions, and secretions [29], the age of donors has been shown to have an impact on the properties and functions of MSCs [30,31], as well as their secretome profiles [32] and potentials [33].

Further understanding of the impact of age and other donor-related factors will be crucial to the development and application of secretome-derived products [15]. To date, age’s effect on MSC potential and secretomes has not been demonstrated [34]; MSCs’ functional properties and factor secretion status were not essentially determined by age, despite their dissimilarity between different human MSC donors and preparations [35].

Another explanation for this possible variation in MSC secretomes among donors is MSC populations. The fraction of ‘‘stem-like’’ cells in a population of MSCs appears to be quite heterogeneous and can vary in proportion depending on the donor (interpopulation heterogeneity). Variability in the secretion of several proteins from cultured MSCs of individual subjects suggests that these cells exist as a heterogeneous population containing functionally distinct subtypes, which differ in numbers between donors [36]. Variation can be found even when the same donor is utilized: a significant difference exists between secretomes of size-sorted MSC subpopulations from the same donor (intrapopulation heterogeneity) [27]. A significantly higher trophic factor-producing capacity is attributed to the large MSC subpopulations [27,37,38]. The majority of factors are pro-osteogenic, pro-senescence, and anti-chondrogenic. Some factors associated with pro-chondrogenic and anti-osteogenic functions are found at a higher level in the small and medium-size subpopulations, and a more significant impact of the size-dependent MSC secretome could be expected in long-term cultures [27]. Current MSC-based clinical trials rarely select for competent subpopulations after culture expansion, which might be an important cause for their inconsistent therapeutic outcomes. Identification and sorting of subpopulations from culture-expanded MSCs may lead to substantial improvements in the therapeutic outcomes, by selecting subpopulations with the most suitable secretome profile for each specific therapeutic application [27].

## 4. Cell Passage Number

Understanding the differences in properties of MSCs at early versus late passage will help refine MSC treatment strategies. Early passage cell populations increase the likelihood of heterogeneity whilst late passage cells retain characteristic markers for MSC phenotype in a selectively more homogeneous population [39]. Moreover, the immunomodulatory properties of MSCs in a long-term culture have been reported [40]. Nevertheless, prolonged in vitro culture of MSCs leads to a changed MSC phenotype and multipotency [30,41,42,43], attenuating their stemness and contributing to reduced therapeutic potential [44]. Stem cells with low passage numbers can secrete larger amounts of therapeutic paracrine factors, which are required for tissue regeneration, as compared to stem cells with high passage numbers [45,46,47,48]. Serra et al. characterized the secretomes obtained from different passages (from passage 3 to passage 12) using proteomic analysis. They revealed that different passages present distinct profiles, with no significant variation in the composition of proteins associated with neuroprotection/differentiation and axonal growth [49].

## 5. Culture Medium

Studies showed that cell culture media could have an impact on the potential of MSCs for adhesion and growth and can be positively selective for specific MSC subpopulations [50,51]. Sagaradze et al. observed that the concentrations of factors are different between MSC-CMs where two different growth media were used [13]. In contrast, Czapla et al. found that MSCs cultured in three different media exhibited similar secretion profiles [52].

Furthermore, some researchers use fetal bovine serum (FBS) in the culture medium for MSC-CM production. Contamination risk from animal proteins is normally present in FBS [53], and thus, immunologic reactions are expected when MSC-CM is used in vivo. Concerns already exist with FBS use such as the variability of FBS from batch to batch [53] due to geographical and seasonal variations [54] and the ill-defined mixture of its components that contains thousands of constituents and can contain contaminants, such as endotoxins, mycoplasma, viruses, or prion proteins [54]. A short culture period may leave some growth factors derived from serum that has not been consumed by the cells [4]. These factors will be added to the growth factors secreted by the cells themselves. Different percentages of FBS result in different amounts of growth factors present in a culture; thus, some MSC manufacturers emphasize the importance of qualifying FBS lots to facilitate product comparability between manufacturing runs [8]. The most common alternative for FBS is human serum and its derivatives such as human platelet lysate [8,55]. However, the effect of the human platelet lysate on the immunomodulatory capacity of MSCs and its reproducibility (donor-to-donor variability) are still contradictory [53]. All these issues make it difficult for MSC secretome-based products prepared in the presence of serum to validate GMP-compliant processes. The most acceptable alternative is serum-free or preferably chemically defined medium, the latter not only serum-free but also lacking any hydrolysates or supplements of unknown composition [55]. Interestingly, it has been shown that serum-deprived cultures of MSCs secreted a higher level of angiogenic factors [56,57]. Also, MSC-CM collected under serum conditions was toxic to cells when used undiluted (100% concentration) and, when diluted, did not have the positive effects of MSC-CM collected under serum deprivation conditions [58].

## 6. Cell Confluency and Conditioning Period

There is significant controversy in the literature regarding the optimal conditions and time points for MSC-CM collection [58]. Cell confluency as well as the period of medium conditioning with MSCs could affect the concentration of secreted factors. The secretome could be richer in factors when there are more cells, or when cells are kept longer in the culture so they become more confluent. However, the expression levels of stemness genes reduce with high cell density [59], with an impact on their secretome. Thus, MSC confluency and conditioning period should be determined carefully before starting an experiment. Mizukami et al. showed that the majority of interesting proteins from the MSC secretome were enriched through time in culture [60]. Sagaradze et al. analyzed the total concentrations of four growth factors in MSC-CM on certain days (until 14 days). Peak factor concentrations were mostly reached on days 7 or 10 for all of them [13]. Most commonly, MSC-CM is collected from 70–90% of MSC confluency during the first three days of culture [4].

## 7. Microenvironment Cues

The use of microenvironment cues to manipulate MSC potency and the MSC secretome in cultures has been extensively explored [61]. They are used to change the MSC secretome profile or to increase growth factor secretion, to enhance the therapeutic capacity/potential of MSC-CM. A variety of different factors were used, in the literature, to change the microenvironment or induce an in vitro preconditioning of MSCs, including the following: 3D culture [62,63,64], hypoxia [65], biochemical stimuli (including cytokines, growth factors, hormones, and pharmacological agents), mechanical stimuli [15,66], electrical stimuli [67], and photobiomodulation [68,69].

Using 3D culture is a more complete modeling of the MSC natural microenvironment, allowing MSC proliferation and differentiation potential to be retained for a longer time [14]. The microenvironment established within the spheroids acts in an autocrine fashion favoring an enhanced secretion of paracrine factors [70]. The composition of the resulting CM is significantly different from that obtained with 2D microenvironments [64,71]. A 3D cell culture is a typical example of microgravity application [72] that significantly increases the anti-apoptotic and anti-inflammatory effects of MSC-CM [73]. Changing the microenvironment in vitro can be produced also via MSC preconditioning. There are different strategies for MSC preconditioning to stimulate growth factor secretion into the culture CM. The application of stress factors, such as serum deprivation or hypoxic conditions, has been widely used for this purpose, to stimulate the stress environment found in damage conditions in the in vivo situation. MSC stimulation with hypoxia triggers an increase in growth factor secretion with enhanced paracrine activity [74,75]. Other factors, treatment with pharmacological molecules or cytokines, and induction of thermal shock have been also applied [58]. For example, we showed in our previous publication that an MSC culture with a B-27 supplement drastically changes the secretome’s profile, further stimulating neurite outgrowth [48]. Also, Park et al. showed that treatment with bFGF and selenium stimulates the production of paracrine factors from MSCs, promoting cell proliferation and migration and the wound-healing process [76]. MSC differentiation affects the secretome profile of MSCs [77]. MSC exposure to mechanical stimuli is another strategy to influence MSC behavior and MSCs’ secretome profile. MSC secretomes respond to the mechanical properties of their substrate, including stiffness [78]. Surface topographies can change the shape of stromal cells and quantitatively influence their cytokine secretion profile. However, qualitative stromal cell secretory characteristics are preserved irrespective of microenvironmental surface factors [79]. A CO_2_ laser enhanced the secretion of pro-angiogenic molecules and increased the regenerative capacity of MSCs [80]. Interestingly, the cell-conditioned medium of an extrinsic microenvironment can modify the age-related properties of tissue-specific stem cells [81].

Each preconditioning regimen induces an individual expression profile with a wide variety of factors, including several growth factors and cytokines [82]. Further studies are required to evaluate the best preconditioning regimen for each specific application of MSC-CM in human regenerative medicine.

## 8. Secretome-Derived Product Purification

The MSC-CM or secretome is primarily prepared by centrifuging the expended medium of MSCs. The resulting product can be used directly, or by adding concentration, fractionation, and/or filtration steps. The preparation procedure of secretome products derived from an MSC culture supernatant differs between studies. It consists of one or a combination of these steps. The centrifugation is essential to remove apoptotic and detached cells, waste tissue material, and cell debris from the supernatant. The centrifugal speed and time are very different between studies. Ultrafiltration technology allows the choice of filtration modules with different molecular weight cut-offs (MWCOs), thus allowing the retention of only parts or the whole secretome [10]. Centrifugal ultra-filter units that have an MWCO of <3 KD are used to retain and concentrate the whole CM [83,84]. MSC-CMs were used concentrated in some studies [85,86] and diluted in others [58,87,88], possibly due to the achievement of the optimum balance between metabolic inhibitory by-products and paracrine stimulatory products [58,89]. Centrifugal ultra-filter units with other pore sizes are used when just a fraction of the CM needs to be retained. By fractionating CM, it is possible to correlate a particular molecular subset or CM fractions with a specific measured effect [15]. Many studies have linked CM paracrine effects to specific molecular size fractions. For example, a CM containing products >1000 kDa was described as having a cardio-protection effect in a murine model of ischemia and reperfusion injury [85]. Protein fractions with a molecular weight in the range of 10kDa-3kDa were the only fraction that could protect neurons against induced neurodegeneration, suggesting that these microproteins could be responsible for the neuroprotection of DPSC-CM [90]. Among four fractions of SHED-CM, only the fraction of <6 kDa could promote neurite outgrowth from dorsal root ganglion neurons [91]. Finally, it is important to note that a filtration with a 0.2 μm pore-size filter is usually performed to remove debris from CM and/or for sterility.

Extracellular vesicles (EVs) represent an important fraction of the cell secretome [92], containing cellular proteins, deoxyribonucleic acid (DNA), ribonucleic acid (RNA), exosomes, and microvesicles (MVs). These EVs are heterogeneous, membranous, cell-derived vesicles with a diameter of approximately 40 to 5000 nm that are released by a variety of cells into their microenvironment [93]. Some confusion still exists in the literature regarding the distinction between exosomes and MVs. The difference between these two terms is based on the vesicle size [94]: exosomes are less than 100 nm (30–100 nm), while microvesicles range from 100 to 1000 nm, but these definitions are flexible as this is still quite a novel research field [95].

The qualitative/quantitative composition, and thus the biological activity of MSC-derived products, is strongly influenced by the isolation method chosen (complete CM, soluble factors, exosomes, or MVs). The isolation of EVs is a challenging procedure. Several methods were introduced and utilized for the isolation and purification of EVs: differential centrifugation/ultracentrifugation with or without sucrose gradient cushion, polymeric precipitation isolation, size exclusion chromatography, immunoaffinity isolation, ultrafiltration and microfiltration technologies, microfluidic devices, and exosome isolation reagents [93,96]. The most widely applied method for concentrating and purifying EVs is isolation by differential centrifugation/ultracentrifugation. It consists of several centrifugations that sequentially increase in speed and time and thus sequentially pellet smaller particles [97]. Ultracentrifugation remains the most commonly used isolation method (81%) [98], even if some authors reported some limitations [99,100,101]. Differential centrifugation/ultracentrifugation only sees recovery rates of up to 25% [102]. There is also evidence that the high forces involved (typically 100,000× *g*) can affect the bioactivity of the EVs themselves [103]. Indeed, each of these isolation methods has advantages and limitations. None of them can offer a high recovery together with high specificity [9,104]; wherefore, 59% of respondents use a combination of methods [98] to achieve a higher quality of EVs. They can be filter-sterilized at the end of the isolation process [12]. Data available in the literature tend to state that secretome (or EV) sterilization is possible using filtration, without apparent loss of efficacy [93]. Figure 2 summarizes the different protocols and techniques followed in the literature to obtain the secretome-derived products used for their regenerative potentials.

EVs have been studied widely in the last few years and are described as key regulators of stem cell paracrine activity [104]. Subsequently, many studies have been conducted to identify whether MSC-CM functions are mainly associated or not with the enriched fractions of EVs. Kumar et al. demonstrated that MSC-CM mediates cardioprotection during myocardial ischemia/reperfusion injury through the exosomes [105]. MVs contributed along with soluble factors to the regenerative effect of MSCs in a study conducted by Ahmed et al. [106]. In contrast, Walter et al. showed that the major paracrine angiogenic effect of MSCs is associated with their soluble factors and not with their EVs [107]. Further studies should be conducted to confirm the benefits of MSC secretome fractionation and determine which fraction of MSC-CM is most effective for each regenerative application, especially since the isolation and characterization of EVs are costly and time-consuming, which impairs their use in clinical practice [104].

The secretome is composed of various elements secreted into the extracellular space. These elements include proteins and extracellular vesicles. These extracellular vesicles can contain apoptotic bodies, which appear during the programmed cell death process [108,109]. The various steps of centrifugation are required to remove these apoptotic bodies.

## 9. Other Manufacturing Conditions

In addition to all the above-mentioned points, the development of a reproducible, scalable, and well-controlled platform is an important key factor in the standardization of the production of MSC-CM and its derivatives. Adherent cell expansion has traditionally been performed on planar surfaces such as well plates and tissue culture flasks for simplicity and easy handling when large numbers of cells are not required. For larger-scale expansion, the transition from planar-based culture to microcarrier-based systems not only allows for higher-density culture (thus reducing the cost of goods) but also for more stringent culture control and monitoring [110]. This also avoids the high risk of contamination due to manual interventions of the manual process. Bioreactors improve the predictability of the composition and function of secretome-derived products. Suspension bioreactors offer a higher level of homogeneity and process control, serving to reduce both batch-to-batch and within-batch variability of cell cultures. Furthermore, bioreactors are used sometimes to provide a specific physiological in vitro environment [111]. Stirred suspension bioreactors are highly scalable, and several variables such as dissolved oxygen, pH, and temperature can be computer-controlled to provide a high level of process control, resulting in more uniform product batches [15].

In addition to manufacturing conditions, storage and transport of MSC secretome-derived products should be considered as they play an important role in maintaining characteristics and functions. It is important to consider the effects of freeze–thaw, stability at various temperatures, and the effects of freeze-drying components [15]. Cells cultured with CM stored at 4 °C before use maintained their viability. However, cell death increases with storage time, and the conservation of CM for more than 2 months decreases the properties that keep the cells viable. This may be due to the degeneration of hormones and growth factors in CM, or a concentration of these factors that is too low for long-term preservation of the target cells [112]. CM can reportedly be stored at −20 °C for several months without experiencing functional deterioration [113]. A study on urinary exosomes showed that freezing at −20 °C resulted in major losses, and freezing at −80 °C enabled almost complete recovery after up to 7 months of storage. Further, protease inhibitors are essential for proper preservation, and extensive vortexing (i.e., 90 s) enabled maximum recovery of thawed exosome samples [114]. Lyophilization is used to guarantee the long-term stability and easy storage and reconstitution of products. This process generates a variety of freezing and drying stresses that can alter the stability of the biological samples [10].

## 10. Future Trends and Recommendations

Cell therapy presents several limitations, such as safety issues, handling difficulties, and elevated cost. Therefore, MSC-CM is a therapeutic alternative, as it brings similar effects to the cells of origin, and it allows for avoiding the issues of cell grafting [115]. Taking into account the possibility to standardize conditioned media, MSC-CM clearly appears as a new therapeutic strategy. The use of such cell-free therapy in regenerative medicine has several advantages over stem-cell-based applications: safety considerations, control of protein content, storage without toxic cryopreservative agents, economical and practical aspects, and the possibility to orientate CM potential by preconditioning [108]. This cell-free approach will guide and promote revascularization and endogenous cell homing [109], with the possibility to specifically orientate tissue regeneration [116]. The use of MSC-CM is locally applied to reproduce stem cells’ intercellular signaling, in order to reproduce the biological effects of the original cells [117]. The topical application of a secretome requires a carrier, such as degradable particles, or can be accomplished using a hydrogel. Hydrogels are colloidal structures that present similitude to extracellular matrices and are fully biocompatible. They appear to be excellent carriers for MSC-CM applications in regenerative medicine [118].

For further clinical applications of MSC-CM, strict controls will be mandatory and enhance the necessity of standardization to produce safe and efficient MSC-CM. Several potential points should be considered: (1) control of total protein concentration and protein profiles and (2) control of key protein presence (according to the targeted therapy). A summary of these controls and potential key proteins is presented in Figure 3.

(1).Total protein concentrations and protein profiles can be assessed and controlled by protein assay kits, such as the Pierce bicinchoninic acid BCA Protein Assay Kit (Thermo Fisher) and the Bradford Protein Assay (Thermo Fisher) [48]. Protein profiles can be controlled by electrophoresis with a bioanalyzer (i.e., Agilent 2100 Bioanalyzer) [116].(2).Specific proteins have to be considered to determine the potential efficiency of MSC-CM for specific applications. For tissue proliferation, FGF and TGF-ß have been described in MSC-CM and could be proposed as interesting candidates [119,120,121,122]. For bone regeneration, the BMP family, IGF, and IGF-BP could be considered [48,119,122]. For angiogenesis, VEGF is an obvious key protein, as it has been described several times in MSC-CM [119,121,122]. For neurogenesis, NGF and BDNF can naturally be used as controls [48,120]. Finally, for clinical applications targeting inflammatory control, proteins involved in the inflammation process could serve as key proteins, such as IL-6, IL-10, and TLR-4 [120,122].

## 11. Conclusions

The effects of MSC-CM are multiple, and even the minimal variability of its composition can strongly affect its activity [14]. Moreover, the process of producing human MSC secretome-derived products is a major consideration in developing standardized criteria to define and qualify the preparation of these products for clinical applications [123]. In this review, we discussed the essential origins of variability in MSC secretome-derived products. Based on literature data and experimental results, to standardize the manufacturing of these products, we recommend the preparation of secretomes from MSCs at a low cell passage number, obtained from relatively young and healthy donors. We also recommend DPSC-CM preparation under serum-free conditions, and collection during the first days of conditioning. Further work should be performed to determine which cell sources, MSC populations, environmental cues, and MSC-CM fractions provide the most potent MSC secretome-based product for each specific application in the field of human regenerative medicine.

## Figures and Tables

**Figure 1 ijms-24-12594-f001:**
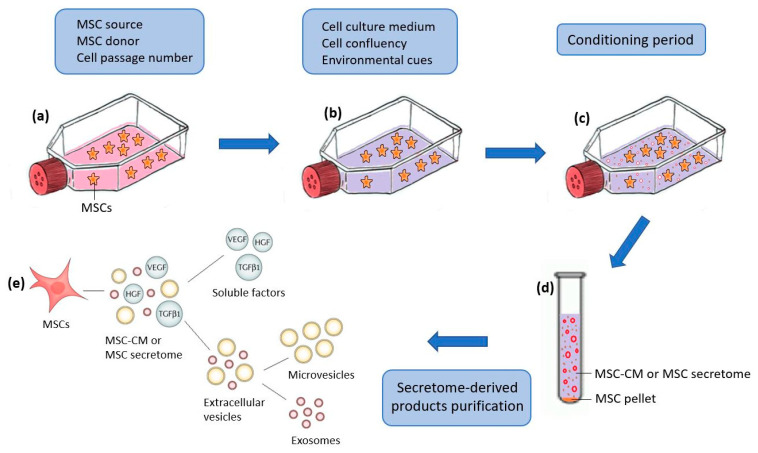
Schematic representation of the procedure, followed in the literature, for obtaining the secretome from mesenchymal stem cells (MSC-CM) and the different variations that can affect its production at the different stages. (**a**) MSC in culture; (**b**) changing the culture medium of adherent cells after reaching a certain degree of confluency; (**c**) MSCs, incubated for some time, release their secretome in the growing medium; (**d**) MSC-CM obtained after collection and centrifugation of the supernatant; (**e**) purification of MSC-CM-derived products.

**Figure 2 ijms-24-12594-f002:**
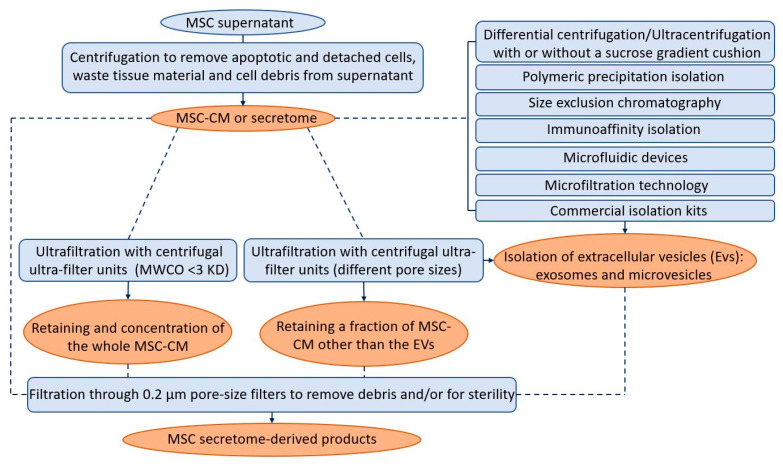
The different techniques (in blue) and protocols (dotted arrows) followed in the literature to obtain the secretome-derived products (in orange) used for their regenerative potential.

**Figure 3 ijms-24-12594-f003:**
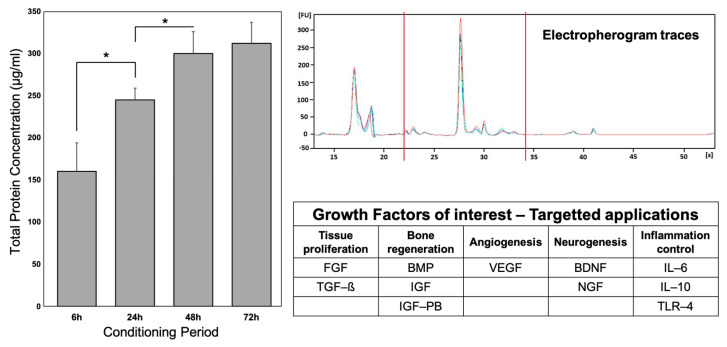
Control of MSC-CM profile and key protein identification. Left graph represents total protein concentrations with BCA Protein Assay kit, according to conditioning period. * indicates significant difference (*p* < 0.05) Right graph represents electropherogram traces of MSC-CM, using Agilent Bioanalyzer. The table summarizes key proteins to consider, according to the therapeutic target. Adapted from [48,116].

## Data Availability

The data presented in this study are available on request from the corresponding author. The raw/processed data required to reproduce these findings cannot be shared at this time due to technical limitations.

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
