# Peer review of "Towards the Standardization of Mesenchymal Stem Cell Secretome-Derived Product Manufacturing for Tissue Regeneration"

_ijms, 2023, doi:10.3390/ijms241612594_

Round 1

Reviewer 1 Report

The present review indicates that the utilization of MSC-CM in clinical settings is currently in progress; however, the outcomes obtained thus far are inconsistent. Consequently, it is prudent to propose a standardized approach for MSC-CM formulation. Nonetheless, achieving such standardization proves to be highly challenging, as the authors have documented numerous conflicting findings. Additionally, they have effectively demonstrated the significant variability prevalent in current research within this particular field. In their concluding remarks, the authors put forth a series of recommendations for future investigations, including the utilization of low passage numbers, serum-free growth conditions obtained from young donors, and early collection of media. However, it would greatly enhance the review if the authors further elaborated on these recommendations. The review identifies several potential areas of focus, such as the inclusion of markers to standardize CM concentration, the identification of key proteins like VEGF, the determination of appropriate application routes (topical or intravenous), the establishment of standardized cell or animal assays to assess effects (specifically, vascularization and epithelialization), and the consideration of cell-to-media ratios or other relevant factors for standardization purposes.

Author Response

Dear Reviewer,

We thank you for your careful evaluation, and valuable comments.

We have modified our manuscript according to your recommendation :

We have added a chapter « Future Trends and Recommendations » in which we develop several potential areas of focus.

All modifications and additions are in red in the MS.

Yours faithfully,

Reviewer 2 Report

The review deals with the important topic of standardisation of secretome and its derived products. The intention is good and correct, highlighting the influence of the production process, source tissue, donors and other issues; however, there is no novelty on the subject compared to other reviews published, including in this journal (2017; 2021; 2022).

In addition to considering that the contribution of new information is scarce, and that experts on the subject such as Vizoso et al. are not cited, the authors make a poor review of the conditions of donors, especially the impact of why they must be young and healthy, and do not define or describe the different particles, leaving aside the apoptotic bodies.

Author Response

Dear Reviewer,

We thank you for your careful evaluation, and valuable comments.

We have substantially modified our manuscript according to your recommendation : the review has been completed with the addition of 10 references, especially from the experts you mentioned.

The impact of apopototic bodies has been included in « 8. Secretome-derived products purification », and chapter « Future Trends and Recommendations » has been added to further discuss the issues raised, according to the content of this review and to previously published works (in particular the added references).

All modifications and additions are in red in the MS.

Yours faithfully,

Reviewer 3 Report

Overall the paper is well-written and very interesting the topic.

Being a review will be nice to add at least another  figure. 

The paper needs to be revised by a native english speaker. 

The paper needs to be revised by a native english speaker. 

Author Response

Dear Reviewer,

We thank you for your careful evaluation, and valuable comments.

We have modified our manuscript according to your recommendation :

We have added a chapter « Future Trends and Recommendations » in which we develop several potential areas of focus. Another figure has been added in this chapter, to summarize some aspects of controls and standardization.

The manuscript has been revised by a native english speaker

All modifications and additions are in red in the MS.

Yours faithfully,

Round 2

Reviewer 2 Report

The authors have responded in part to the comments but I still feel that there is little new information provided.